# Models for the Economic Impacts of Forest Disturbances: A Systematic Review

**Jun Zhai \* and Zhuo Ning**

College of Economics and Management, Nanjing Forestry University, Nanjing 210037, China
\* Correspondence: zhaij@njfu.edu.cn

**Abstract:** The frequency of forest disturbances has increased under intensifying climate changes, and the economic impacts of forest disturbances should receive more attention. This paper systematically reviews models of the economic impacts of forest disturbances and summarizes four commonly applied models, which are "with and without" analysis, equilibrium models, the intervention model, and the social welfare model. The advantages and disadvantages of these four models are presented and compared, and literature using these models is introduced. The review of the economic assessment of damage caused by forest disturbances is expected to provide insights for researchers in this field and useful information about economic losses, price dynamics, and welfare redistribution for policymakers.

**Keywords:** forest disturbances; "with and without" analysis; equilibrium model; intervention model; social welfare model

## 1. Introduction

Forest disturbances such as wildfires, insect infestation, and hurricanes are an integral part of nature that could result in significant changes in the forest structure and fluctuations in the supply of goods and services in the forest ecosystem [1–3]. The frequency and intensity of forest disturbances are increased by the intensification of climate change, and climate change deteriorates society through damage to the services that forest ecosystems provide [1,4–8]. Previous analysis has addressed the ecological consequences of forest disturbances in different regions, but the economic impacts of forest disturbances are not systematic, and a review of economic modelling is still missing to date. The review of models of the economic impacts of forest disturbances is important because even a single insect infestation could cause damage of up to tens of millions of dollars [9,10].

Salvage logging is a form of logging that removes trees and other biological material from sites because of their economic value after natural disturbances [11]. Forest disturbances, along with the following salvage loggings, could severely affect the timber market and economic activities both in the short and long run through the supply of forest products. For example, the Biscuit Fire affected nearly 4 billion board feet (BBF) of softwood tress, and the amount of salvageable timber was estimated to be 0.83–2 BBF, whereas the amount of actual salvage timber was 60 million board feet [12]. The damaged volume from the Biscuit Fire has a positive effect on the timber price during salvage logging and a negative effect after salvage logging [13].

The review of economic methodologies is of vital importance because it could provide potential directions for future research. Depending on the objectives and the techniques of researchers, different models could be determined. In addition, the understanding of the economic impact of forest disturbances is necessary to safeguard the stable production of goods and services and increase social welfare. Economists have developed and employed various economic methodologies to assess the impacts of forest disturbances on economic activities and provide suggestions for policymakers to prevent or reduce the economic consequences of forest disturbances.

Typical economic methodologies to study the economic impacts of forest disturbances include "with and without" analysis, equilibrium models, the intervention model, and the social welfare model [9,13–22]. "With and without" analysis is the most straightforward and primary estimation of the economic losses caused by forest disturbances. The equilibrium models consider the impact of forest disturbance on a specified sector or the whole economic sector. The intervention model focuses on the dynamic impacts during different stages of the forest disturbance on timber prices and distinguishes between producers with damaged timbers and producers with undamaged timbers. The social welfare model estimates the welfare gains and losses for consumers, producers with damaged forests, and producers for undamaged forests. Each of these models has its own focus and advantages; however, how these models distinguish from others are not identified, and the applications are not systematic reviewed.

Therefore, the objective of this literature review is to highlight the most relevant papers that investigate the economic impacts of forest disturbances to provide researchers with the most comprehensive understanding of methodologies for modeling the economic impacts of forest disturbances. The damage appraisal after forest disturbances is important for forest managers and policymakers to assess whether it is economically efficient to adapt different rescue and salvage programs. This economic modeling review is beneficial for both forest managers at the microeconomic level and policymakers at the macroeconomic level. The forest managers need to take account of timber prices that are affected by forest disturbance in forest growth and decide when to harvest or when to salvage when the forest disturbances occur. The policymakers need to understand the direct and indirect impacts from forest disturbances on the economy sectors and make the optimal management actions.

## 2. Materials and Methods

To explore the modeling of the economic impacts of forest disturbances, we exploited the Web of Science (WOS) database for relevant studies at the global scale and conducted a literature search for papers that were published in English only. The forest disturbance categories considered in this research include (1) wildfire, (2) pest, (3) pathogen, (4) storm, (5) snow, (6) others. The articles were collected in May 2022, and the time range is not predefined. Each search includes three keywords as follows: Forest *AND Economic* AND disturbances in Table 1. A total of 963 papers in the field of economic impacts of forest disturbances are filtered, and 100 articles are selected based on their relevance. Within the 100 articles, only 25 are the highly relevant papers that model the economic impacts of forest disturbance on the market. The article selection workflow of 963 papers is presented in Figure 1.

**Table 1.** Keywords per disturbance.

| | Disturbance Category | Keywords |
|---|---|---|
| (1) | Wildfire | fire OR wildfire OR forest fire OR bush fire |
| (2) | Pest | pest OR infestation OR insect |
| (3) | Pathogen | pathogen OR infection OR disease |
| (4) | Storm | storm OR hurricane OR tornado OR cyclone OR typhoon OR wind OR windstorm |
| (5) | Snow | snow OR ice |
| (6) | Other abiotic disturbances | drought OR flood OR landslide OR volcano |

The distribution of publication by journal is presented and discussed. Table 2 presents the breakdown of articles by journal. As can be seen from Table 2, 5 out of 25 papers were published in *Forest Policy and Economics*, 4 are published on the *Canadian Journal of Forest Research*, and 3 on *Forest Science.* Two papers are published in journals including *Environmental Management*, *Ecological Economics*, *Forests*, and *Journal of Forest Economics*. Only one paper was published in other journals. The top three journals are forestry journals with high reputations in the field of forest economics.

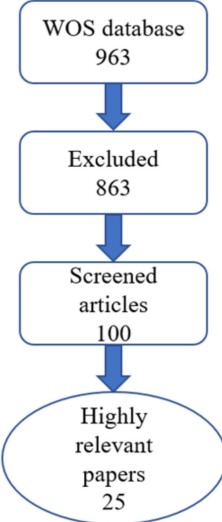

**Figure 1.** Article selection workflow of 963 papers resulting from the WOS query.

**Table 2.** Overview of the main scientific journals.

| Journal Title | Number of Papers |
|---|---|
| *Forest Policy and Economics* | 5 |
| *Canadian Journal of Forest Research* | 4 |
| *Forest Science* | 3 |
| *Environmental Management* | 2 |
| *Ecological Economics* | 2 |
| *Forests* | 2 |
| *Journal of Forest Economics* | 2 |
| *Journal of Forestry* | 1 |
| *Forestry: An International Journal of Forest Research* | 1 |
| *Journal of Forestry* | 1 |
| *Agricultural and Resource Economics Review* | 1 |
| *Journal of Environmental Management* | 1 |

## 3. Models

A forest disturbance is defined by the FAO as "Damage caused by any factor (biotic or abiotic) that adversely affects the vigor and productivity of the forest and which is not a direct result of human activities" [23]. From the perspective of economics, a forest disturbance is "an event that interrupts or impedes the flow of goods and services provided by forest ecosystems that are desired by people" [24]. The commonly applied models in assessing the economic impacts of forest disturbances are "with and without" analysis, general or partial equilibrium models, the intervention model, and the social welfare model. Depending on the objective, the availability of data, and the tolerance of limitation, these four models can be applied differently. The literature using these models is presented in the subsections below.

### 3.1. The "with and without" Analysis for Economic Loss Evaluation

The economic impact can be evaluated through two situations, the first situation being without the forest disturbances, and the second situation being with the forest disturbances. This approach is called the "with and without" analysis. The difference between the present value of the forest resource with and without forest disturbances is the traditional damage appraisal technique [25–27]. The assessment of economic losses provides useful information for management program decisions. Citing the elasticities of the corresponding markets and market prices, the potential impacts of forest disturbances on welfare could be

quantified. This estimation of market welfare and the measurement of welfare distribution among participants provide important references for policymakers.

Although the essence of the "with and without" analysis is economic loss evaluation, the loss could be the loss value, removal cost, discounted cost of all expenses, and economic loss. Therefore, the techniques for damage appraisal after the forest disturbances are different depending on the estimated loss. As displayed in Table 3, the commonly applied techniques to estimate the economic losses include net present value and Monte Carlo simulation. The literature estimating the various economic losses, along with their techniques, is presented in Table 3.

**Table 3.** Summaries of literature using the theoretical framework.

| No. | Author (Year) | Forest Disturbance | Technique | Main Findings |
|---|---|---|---|---|
| 1 | Michalson (1975) | MPB in Idaho | Demand model | A capitalized loss value of approximately USD 4.7 million for the current infestation situation, USD 7.6 million for the infestation of all Targhee campgrounds, and USD 3.8 million for the infestation of half Targhee campgrounds [28]. |
| 2 | Guimaraes et al. (1993) | Hurricane Hugo in South Carolina | Net present value | The direct economic losses of USD 1.3 billion in lost crops and timber, USD 3.0 billion in residence, and USD 1.0 billion in commercial and industrial structures [29]. |
| 3 | Butry et al. (2001) | Wildfires in Florida | Net present value | A net economic loss of at least USD 600 million with quantified losses for undamaged forest owners, damaged forest owners, and consumers [30]. |
| 4 | Haight et al. (2011) | Oak wilt in Minnesota | Landscape model | The death of 76–266 thousand trees will be infected with the removal cost of USD 18–60 million [18]. |
| 5 | Kovacs et al. (2011) | SOD in California | Simulation | The discounted cost of the treatment, removal, and replacement of more than 10 thousand oak trees is USD 7.5 million, and the discounted property value loss is USD 135 million [19]. |
| 6 | Pye et al. (2011) | SPB in the southern United States | Net present value | The short-run annual economic losses of forest landowners is USD 43 million, and that of wood-product consumers is USD 30 million [20]. |
| 7 | Zhao et al. (2020) | Pine wood nematode disease in China | Market value model | The average annual economic loss of CNY 7.17 billion, of which the direct loss is CNY 1.53 billion and the indirect loss is CNY 5.64 billion [31]. |
| 8 | Knoke et al. (2021) | Natural disturbances in Norway | Monte Carlo simulation | The economic losses induced by natural disturbances range from -€2611 to -€34,416 per hectare depending on the chosen evaluation approach [32]. |

Michalson (1975) estimated the economic impact of MPB by subtracting the economic value with MPB from that without MPB in Idaho's Targhee National Forest [28]. Approximately 500 recreational users in six campgrounds were interviewed during July and August of 1973. The number of visitor days per trip is regressed on the round-trip mileage, estimated travel time, and cost per visitor day using multiple regression least squares analysis. The economic value of outdoor recreation value is calculated, and the present values are determined. Their results indicated that an annual economic loss of more than USD 500,000 would be caused if all campgrounds were infected with MPB. They found a capitalized loss value of approximately USD 4.7 million for the current infestation situation, USD 7.6 million for the infestation of all Targhee campgrounds, and USD 3.8 million for the infestation of half Targhee campgrounds.

Guimaraes et al. (1993) take the example of Hurricane Hugo to investigate the wealth and income effects of natural disasters [29]. Hurricane Hugo attacked South Carolina in September 1989, causing 25 deaths, and 24 out of 46 counties were declared disaster areas [29]. A multi-sector regional econometric model is employed to estimate the effect of Hurricane Hugo on the state's economy through the "with and without" analysis. They estimate direct economic losses of USD 1.3 billion in lost crops and timber, USD 3.0 billion in residence, and USD 1.0 billion in commercial and industrial structures, whereas the reimbursements from federal flood insurance and private insurance claims for structural

damage was USD 2.891 billion [29]. Out of the USD 1.2 billion damage to timber, only USD 150 million was salvageable [29].

The catastrophic wildfires in northeastern Florida in June and July 1998 burned an area of approximately 500,000 acres of forest and caused an economic loss of at least USD 600 million, which was close to the loss from a category-2 hurricane [30]. In the study by Butry et al. (2001), seven major categories of costs and losses were quantified and examined, which included pre-suppression costs, suppression costs, disaster relief expenditures, timber losses, property damages, tourism-related losses, and human health effects and economic effects on forestland owners and consumers. Using a discount rate of 6%, forest landowners who experienced no wildfire losses were estimated to gain total welfare between USD 400 million to USD 1.4 billion in the long run because of the higher price, whereas forest landowners experienced with wildfire damages gained a total welfare between USD 33 to USD 61 million from salvage revenues. However, consumers were expected to lose a total welfare between USD 400 million to USD 1.5 billion in the long run because of the higher log price, though they benefited from the price discounts and greater volume from salvage loggings in the short run.

Haight et al. (2011) predict the economic impact of oak wilt in Anoka County, Minnesota, for the period of 2007–2016 [18]. The invasive alien pathogen *Ceratocystis fagacearum* is a fungus that attacks oaks (*Quercus spp.*) and results in oak wilt in the central United States. A landscape-level model is developed for oak wilt to predict the economic loss from oak wilt mortality in the absence of management. Based on the estimation of 5.92 million oak trees and 885 active oak wilt pockets in 2007, they predict that 76–266 thousand trees will be infected with the removal cost of USD 18–60 million. Therefore, this study suggests that the economic benefits are significant if the oak wilt is prevented and controlled.

Kovacs et al. (2011) employ the spatio-temporal stochastic epidemiological model for the sudden oak death (SOD) infestation spread to predict the economic costs and property value losses from sudden oak death in the period of 2010–2020 in California [19]. The sudden oak death (SOD), *Phytophthora ramorum*, is an invasive forest pathogen that causes substantial mortality in coastal live oak and several other oak tree species on the Pacific Coast of the United States. They predict that the discounted cost of treatment, removal, and replacement of more than 10 thousand oak trees is USD 7.5 million, and the discounted property value loss is USD 135 million.

The economic impacts of SPB outbreaks on timber growers and wood-product consumers from 1997 to 2004 in the southern United States was also studied by Pye et al. (2011) [20]. A theoretical framework linked with a model of the timber market impacts of SPB was used to estimate the impacts on timber producers and wood-using firms separately. They estimated the short-run annual economic losses of forest landowners to be USD 43 million and that of wood-product consumers to be USD 30 million. To mitigate the negative impacts on forest landowners from depressed prices, they suggested that forest landowners should forego timber harvests during the stage of salvage logging, weight restrictions on the road should be adjusted to facilitate salvage logging on private forestlands, and damaged timber on public forestlands should be kept out of the market.

Zhao et al. (2020) quantitively evaluated the direct economic losses from pine wood nematode disease in mainland China from 1998 to 2017 at the provincial level using the market value method [31]. Pine wood nematode disease was introduced to Nanjing, China in 1992 and rapidly spread to surrounding provinces. The direct economic losses include a loss of forest material resources, loss of prevention and control expenditures, and ineffective forestry expenditure. Their results suggest that the average annual economic loss is CNY 7.17 billion, of which the direct loss is CNY 1.53 billion. Zhejiang Province, Guangdong Province, and Jiangsu Province are the most affected provinces, with annual economic losses of CNY 2.14 billion (26.8%), CNY 1.81 billion (22.7%), and CNY 1.22 billion (15.3%), respectively.

Knoke et al. (2021) developed a new methodological approach, the empirical function of forest growth with survival model, for Norway spruce forests and use Monte Carlo simulations to quantify the economic losses from natural disturbances [32]. The rate of natural

disturbances is assumed with a prior probability, and tree survival and mortality rates are cited from other research. Through the simulation, they find that the economic losses induced by natural disturbances range from −€2611 to −€34,416 per hectare depending on the chosen evaluation approach. In addition, they find that the extreme disturbance events increase the economic losses from 50% to 95% compared with not considering these events.

### 3.2. Equilibrium Models

The general equilibrium (GE) framework model allows the policymakers to account for all stands and transactions in the economy because every sector is linked to other sectors directly through transactions or indirectly through the competition for labor, capital, and land in the production stage [33,34]. The assessment of the potential impacts of forest disturbances in a general equilibrium model could assist policymakers with the design of policies for regional development. In equilibrium, a set of equations are expressed for prices and production levels in each sector, and as a result, the total supply of commodities equals the total demand.

The computable general equilibrium (CGE) model is the general equilibrium model with flexible prices and has been utilized by researchers to examine the economic impacts of timber supply shocks from forest disturbances on the industrial sector [17,21,34]. The partial equilibrium model only includes a specific sector, and it includes models such as the global forest product model (GFPM) and the French Forest Sector Model (FFSM). The GFPM is a dynamic economic equilibrium model for the world forest sector, which is an effective model for international environmental issues in forestry [35]. The GFPM is a partial equilibrium model because it only includes the forest industry, with other sectors indirectly accounted for. The French Forest Sector Model (FFSM) is a dynamic partial equilibrium model of the French Forest Sector. The literature that has applied the CGE models and partial equilibrium models to evaluate the economic impacts of forest disturbances is listed below in Table 4.

**Table 4.** Summaries of articles using equilibrium models.

| No. | Author (Year) | Forest Disturbance | Region | Main Findings |
|---|---|---|---|---|
| 1 | Patriquin et al. (2007) | Mountain pine beetle (MBP) | British Columbia, Canada | A relative boom to regional economies in the short run but a negative impact in the long run [17]. |
| 2 | Abbott et al. (2009) | MBP | British Columbia, Canada | The MBP outbreak would increase the prices of sawlogs by USD 12/m$^3$ (13%) in British Columbia [36]. |
| 3 | Chang et al. (2012) | Spruce budworm (SBW) | New Brunswick, Canada | The economic output in New Brunswick from 2012 to 2041 will decrease by CAD 3.3 billion under the moderate outbreak and CAD 4.7 billion under the severe outbreak [21]. |
| 4 | Caurla et al. (2015) | Hurricane Klaus | southwestern France | The compensation plan of subsidies within 6 weeks after the storm increases the windfall supply by 14% [37]. |
| 5 | Corbett et al. (2016) | Mountain pine beetle (MBP) | British Columbia, Canada | The GDP is expected to reduce by CAD 57.37 billion (1.34%) and the net welfare is expected to decrease by CAD 90 million from 2009 to 2054 [38]. |
| 6 | Boccanfuso et al. (2018) | Forest disturbances | Quebec, Canada | The gross domestic product (GDP) of CAD 300 million (0.12%) over the 40 years for the Quebec economy [39]. |
| 7 | Petucco et al. (2020) | Ash dieback | France | The prices rise immediately following the negative supply shocks, especially in the Northeast microregion [22]. |

The BC Ministry of Forests and Range predicted that the volume of log harvested in 2006 would be $8.7 \times 10^6$ m$^3$ higher than the annual allowable cut before the outbreak, and the annual allowable cut would be $12 \times 10^6$ m$^3$ lower than the cut before the outbreak. Using a CGE model, Patriquin et al. (2007) investigate the regional economic impact sensitivity to the current mountain pine beetle (MPB) outbreak in five study areas of British Columbia, Canada [17]. The results discovered a relative boom to regional economies in

the short run because of the increased volume of available timber from the MPB but a negative impact in the long run because of the exception that the timber supply fell below the baseline level. However, Patriquin et al. (2007) suggested that the accuracy of estimates could be improved by more flexible functional forms and econometric estimations instead of the Cobb–Douglas production and consumption functions [17].

Abbott et al. (2009) study the effect of the MPB outbreak on log prices in British Columbia using the dynamic global forest products model (GFPM) [36]. They find that the decrease in the log supply due to the Mountain pine beetle would increase the prices of sawlogs by CAD $12/m^3$ (13%) in BC's interior. The outbreak of the MPB in British Columbia would also increase the log price and the lumber price in the southern United States. Additionally, the timber shortage induced by the MPB outbreak in BC would lead to an increase in lumber production in Japan.

The spruce budworm (SBW) is a forest insect and a natural disturbance that affects the large-scale mortality in balsam fir. The SBW outbreak in the 1970s and 1980s caused moderate-to-severe defoliation (30–100% of foliage in the current year) and affected 3.5 million acres of forestlands in the 1970s and 2.2 million acres in the 1980s [21]. Because the SBW outbreak was predicted to occur every 35 years [40], Chang et al. (2012) investigated the potential economic impacts of future SBW on 2.8 million hectares of Crown land in New Brunswick, Canada, over 2012 to 2041 using the dynamic CGE model [21]. Sixteen scenarios are assumed in the model, which include two pest outbreak severities (moderate versus severe), four control program levels (0%, 10%, 20%, and 40% of susceptible area), and two pest management strategies (without versus with replanting harvest and salvage logging schedules). They find that the economic output in New Brunswick over the 30 years would decrease by CAD 3.3 billion under the moderate outbreak and CAD 4.7 billion under the severe outbreak.

Hurricane Klaus attacked southwestern France on 24 January 2009 and caused 42 million $m^3$ of damaged trees. Subsequently, a bark beetle attack in 2011 increased the damaged wood by 7 million $m^3$. As a result, the French government launched a compensation plan of €138.5 million (M) for subsidized rate loans (€12.5 M), storage area subsidies (€25 M), transport subsidies (€56 M), and transshipment subsidies (€46 M) within 6 weeks after the storm. Caurla et al. (2015) modeled the economic impacts of the compensation plan after Hurricane Klaus using the French Forest Sector Model (FFSM), a partial equilibrium economic model of the French forest sector, and compared this plan to alternative plans [37]. Their results indicate that the compensation plan increased the windfall supply by 14%. In addition, the storage-oriented compensation plan could have reduced the fall in price, increased the storage volume, and increased the total gains of overall sectors.

Corbett et al. (2016) estimated the impacts of the mountain pine beetle (MPB) infestation in British Columbia on provinces' future economic prospects by examining the effects of MPB on timber supply reduction from 2009 to 2054 [38]. Using the dynamic computable general equilibrium (CGE) model, the total value loss is predicted to reduce of gross domestic product (GDP) by CAD 57.37 billion (1.34%), and the net welfare is predicted to decrease by CAD 90 million. They expect policymakers to take this research as a reference when they are deciding on policies over the control and management of MPB.

Boccanfuso et al. (2018) proposed that climate change impact forests through natural disturbances such as drought, forest fires and infestations and analyzed the economic impact of climate change on the forest industry in Quebec, Canada [39]. A recursive dynamic computable equilibrium model (CGE) was applied for a period of 40 years. Their results indicate that natural disturbances induced by climate change decreased the gross domestic product (GDP) by CAD 300 million (0.12%) over the 40 years for Quebec's economy. In addition, the forestry sector suffers the most, with losses ranging from 3% to 7.5% for the forestry, wood products, and sawmill and wood preservation sectors.

The challenging aspect of pathogen evaluation is that pathogens not only reduce the current availability of affected species but also the future availability. Petucco et al. (2020) evaluated the economic impacts (resource impacts, market impacts, and welfare impacts)

of an invasive forest pathogen in France, ash dieback (Hymenoscyphus fraxineus), using the French Forest Sector Model (FFSM) [22]. Petucco et al. (2020) predicted that the share of ash in the northeastern landscape will decrease from 7% to 1.5% during the period of 2008–2060 [22]. Moreover, they found that prices rose immediately following the negative supply shocks, especially in the Northeast microregion, but this increased price would not trigger the increase in ash harvest. Finally, their results indicate that the producer surplus for pulp and fuelwood decreases rapidly in the Northeast microregion but increases in the Centre and Southwest.

*3.3. Intervention Model*

The intervention model is a market model that allows for the examination of the impact of various disturbances on the long run effect of an intervention. The advantages of the invention model over the traditional damage appraisal approach are twofold. First, the traditional damage appraisal technique only considers the difference with and without the damage, but it does not account for salvage logging nor the impact of salvage logging on the market's equilibrium price. The welfare estimates without considering the depressing effect of salvage logging on market prices might be inaccurate, and the policy implications might be misleading. Second, the producers are not distinguished into producers with damaged forests and producers with undamaged forests in the traditional method. There exists a welfare transfer between forest landowners with damaged forests and without damaged forests. Ignoring the difference in producers and welfare transfers between producers result in a less comprehensive presentation of the impact of forest disturbances.

The price fluctuations after forest disturbances depend on: (1) the amount of damaged timber; (2) the possibility to increase exports and decrease imports; (3) the possibility to store roundwood; (4) the utilization capacity of the local wood-processing industry; (5) the price elasticity of demand; (6) the quality of salvage logging [37]. Because of the difference in these aspects, the timber price dropped with different intensities and durations after different forest disturbances [41–43].

Following Prestemon and Holmes (2008) [44], the log price dynamics affected by the price elasticity of demand during different stages of the forest disturbance are illustrated in Figure 2. The demand curve is $D$, and the supply curve before the forest disturbance is $S_0$, which generate an equilibrium price of $P_0$ and an equilibrium quantity of $Q_0$ at point $a$. When the forest disturbance happens, the available inventory for harvest deceased, and the supply curve shifts upward to $S_1$. The new log supply curve $S_1$ intersect with the demand curve $D$ at point $b$, generating a higher market equilibrium log price of $P_1$ and a lower market equilibrium quantity of $Q_1$. When there is a salvage logging, a vertical supply curve of $V_0$ is introduced. The salvage supply curve is vertical because forest landowners are willing to take any price greater than zero. Then the volume of salvage logging drives the supply curve outward to $S_2$, and the new supply curve intersects with the demand curve $D$ at point $c$. A lower market log price of $P_2$ and a larger quantity of $Q_2$ are generated during salvage logging. After the salvage logging is finished, the salvage logging supply curve $V_0$ disappears, and the market supply curve of logs shifts from $S_2$ to $S_3$. The new supply curve $S_3$ intersect with the demand curve $D$, generating a higher log price than that before the disturbance of $P_3$ and a lower quantity of logs supplied $Q_3$. The research that has applied the invention model to study the impacts of forest disturbances on timber price dynamics are presented in Table 5.

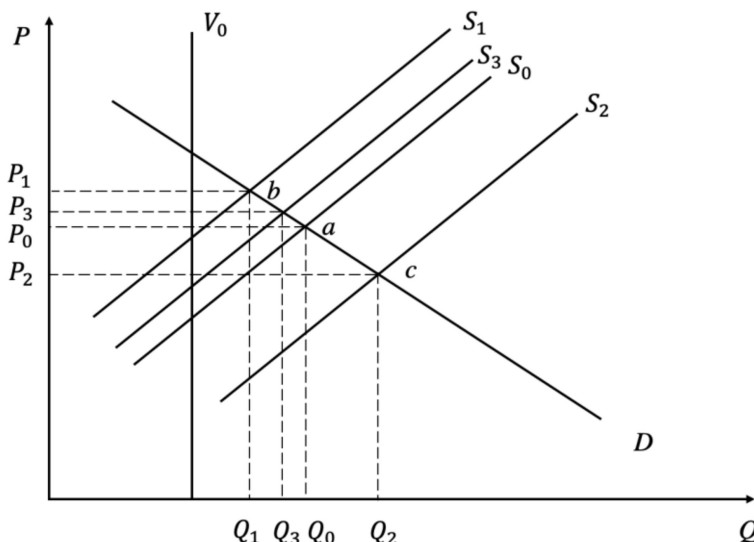

**Figure 2.** Market supply and demand shifts during different stages of salvage logging.

**Table 5.** Summaries of literature using the intervention model.

| No. | Author (Year) | Forest Disturbance | Region | Main Findings |
|---|---|---|---|---|
| 1 | Holmes (1991) | Southern Pine Beetle (SPB) | Texas; Louisiana | The stumpage price of southern yellow pine sawtimber decreased by USD 34.82 in Texas and USD 34.52 in Louisiana [9]. |
| 2 | Yin and Newman (1999) | Hurricane Hugo | South Carolina | A slightly lower price in the short run but no persistent higher price in the long run for hardwood sawtimber and pine pulpwood markets [14]. |
| 3 | Prestemon and Holmes (2000) | Hurricane Hugo | South Carolina | The timber price dropped by 30% in the short run due to the salvage logging but increased by 10% to 30% in the long run [15]. |
| 4 | Baade et al. (2007) | Hurricane Andrew | Miami | A 2.29% decrease in taxable sales in the immediate aftermath of a 5.53% increase by the following month, and reach the previous level in the long run [45]. |
| 5 | Zhai and Kuusela (2020) | The Biscuit Fire | Oregon, USA | A positive effect on log prices in the short run, and a negative effect in the long run [13]. |

The Southern Pine Beetle (SPB) (*Dendroctonus frontalis*) is the most destructive insect infestation to southern pine forests in the southern United States, and the value of trees killed by SPB amounted to USD 1.5 billion from 1970 to 1996 [46]. Recently, the largest SPB outbreak happened from 2000 to 2002 and caused a mortality of trees with a volume over a billion cubic feet [18]. Because the traditional technique for forest damage appraisal failed to consider the lower equilibrium log price because of the salvage logging, the intervention analysis was introduced to develop a new method to estimate the impacts caused by catastrophic SPB in the Texas–Louisiana epidemic in the short run [9]. The intervention analysis was introduced by Box and Tiao (1975) [47] to study the effect of interventions on given response variables and has been widely applied to economic and environmental problems. Holmes (1991) found that the stumpage price of southern yellow pine sawtimber decreased by USD 34.82 in Texas and USD 34.52 in Louisiana [9]. The total welfare loss was estimated to range from USD 31.7 million to USD 78.2 million in Texas and from USD 9.7 million to USD 24.4 million in Louisiana.

Yin and Newman (1999) studied the effect of Hurricane Hugo on sawtimber and pulpwood stumpage prices for both softwood and hardwood in the coastal plain region of South Carolina [14]. An intervention analysis was used to analyze the quarterly data

from the 1st quarter of 1977 to the 4th quarter of 1996. They found a slightly lower price in the short run but no persistent higher price in the long run for hardwood sawtimber and pine pulpwood markets. No significant price drop in the hardwood pulpwood market immediately after the Hurricane Hugo was discovered, but a higher price of USD 6/MBF in the long run was detected in the pine sawtimber market. Prestemon and Holmes (2000) also investigated the short-run and long-run effects of Hurricane Hugo on timber prices using an intervention model [15]. They found that the timber price dropped by 30% in the short run due to salvage logging but increased by 10% to 30% in the long run.

Hurricane Andrew in August 1992 in Miami was the most expensive natural disaster in U.S. history before Hurricane Katrina, with property losses of over USD 26 billion. Using the intervention analysis on an ARIMA model, Baade et al. (2007) assessed the economic impact of Hurricane Andrew on local, regional, and national economies and provided lessons for the economy recovery after Hurricane Katrina [45]. Their results indicate that the taxable sales in south Florida fell by 2.99% in August 1992 but rebounded to a level of 5.53% above the usual level in September 1992, and returned to their previous levels over the following 18 months. In addition, their results suggested an immediate drop in taxable sales of USD 198 million in August 1992, the peak point within 3 months, and a bump in the economic activity for the following 18 months.

Zhai and Kuusela (2020) go further and investigated the price dynamics in three regional log markets in Oregon after the Biscuit Fire by dividing the short run into the period immediately after the Biscuit Fire and the period of salvage logging [13]. The period immediately affected by the Biscuit Fire was in the third quarter of 2002, the period of the short run effect ranged from the fourth quarter of 2002 to the fourth quarter of 2006, and the period of the long run effect started from the first quarter of 2007. A univariate intervention model and a reduced form model were applied to analyze data from the second quarter of 1993 to the first quarter of 2017. They found no significant immediate effect in the first quarter after the Biscuit Fire in any of the models, a positive effect on log prices during salvage logging in some markets, and a negative effect after salvage logging in some markets. They explain that the reason for the positive effect during salvage logging is that the small volume of salvage logging is not enough to reduce log prices in the regional markets. In addition, they explain that the reason for the negative effect after salvage logging is that the prices capture some exogenous variables, such as the Great Recession.

*3.4. Social Welfare Model*

One of the most concerned aspects of human welfare by economists is economic surplus, which is the difference between the benefits and costs of producing goods and services. Salvaged timber is an importance source of timber. The social welfare model quantifies the welfare impacts from salvage logging on different market participants and timber markets. Especially, the market distributional consequences of salvaging among forest landowners with damaged forests (including losses from damaged trees and gains from salvage logging), forest landowners with undamaged forests, and consumers could be specified. As a result, the application of the social welfare model could have important implications for governments regarding the mitigation of economic losses through salvage logging. The literature using the social welfare model to estimate the welfare distribution among market participants is presented in Table 6.

**Table 6.** Summaries of literature using the social welfare model.

| No. | Author (Year) | Forest Disturbance | Technique | Main Findings |
|---|---|---|---|---|
| 1 | Prestemon and Holmes (2004) | Hurricane Hugo | South Carolina | Consumers gained a welfare of USD 5.4 million for each percent of timber salvaged, producers of damaged timber gained a welfare benefit of USD 6.4 million, and producers of undamaged timber lost a welfare of USD 5.6 million [48]. |
| 2 | Prestemon et al. (2006) | Fires | Bitterroot National Forest | The mediation plan of salvaging the volume of 0.27 million m$^3$ reduced revenues from salvaging to the U.S. treasury by USD 8.5 million and caused a net welfare loss of USD 8.8 million [16]. |
| 3 | Holmes et al. (2008) | Biscuit Fire | Oregon | The producer of damaged timber lost a welfare of USD 51.5 million, whereas the consumer lost a welfare of USD 79.7 million [24]. |
| 4 | Prestemon and Holmes (2010) | Six severe hurricanes | the United States | The economic estimates of these six severe hurricanes are quantified, and suggestions are proposed [49]. |
| 5 | Chang et al. (2011) | Spruce budworm (SBW) and forest tent caterpillar (FTC) | Two Canadian provinces | A welfare gain of CAD 14.3–20.8 million for SBW control in NB, CAD 7.9–14.5 million for FTC control in NB, CAD 22.2–32.4 million for SBW control in SK, CAD 11.7–22.0 million for FTC control in NB [50]. |
| 6 | Henderson et al. (2022) | Hurricane Michael | Florida | The welfare gain for pine sawtimber producers increases by 1.2 to 1.5 times, and the welfare for consumer decreases by 0.6 to 0.8 times [51]. |

Prestemon and Holmes (2004) focused on Hurricane Hugo in South Carolina and quantified the market dynamics of short-run and long-run welfare impacts [48]. They divided the market participants into producer surplus gained from salvaging for owners of damaged timber, producer surplus lost from killed timber for owners of damaged timber, producer surplus for owners of undamaged timber, and consumer surplus. The welfare loss for consumers is the long-run welfare impact because it includes the gain during salvage logging and the losses after salvage logging. They find that consumers gained a welfare of USD 5.4 million for each percent of timber salvaged, producers of damaged timber gained a welfare benefit of USD 6.4 million, and producers of undamaged timber lost a welfare of USD 5.6 million.

The fires in the Bitterroot National Forest in the northern Rocky Mountains that occurred in the summer of 2000 burned an area of 124,520 hectares and caused damages to timber of 5.33 million m$^3$ [16]. A plan of salvaging 15% of the burned area, which is 0.8 million m$^3$ of damaged timber, was proposed but challenged in court. Instead, a mediation plan of salvaging a volume of 0.27 million m$^3$ (60 MMBF) was approved but delayed the initiation to 2003. Prestemon et al. (2006) quantified the economic costs due to the decreased salvage volume and the decay from administrative delays by calculating the welfare changes for consumers, producers, and the treasury [16]. They find that the mediation plan reduced revenues from salvage to the U.S. treasury by USD 8.5 million and caused a net welfare loss of USD 8.8 million. Additionally, they found that the administrative delay of salvage logging decreased revenues from salvage logging to the U.S. treasury by USD 1.5 million (25%).

The catastrophic Biscuit Fire happened between 13 July and 9 November 2002, burning a forest area of 499,965 acres, mainly on the Siskiyou National Forest, and killing a volume of softwood timber of 1619 million board feet (MMBF). The degrade factor is the weighted average of deterioration rates by species [24]. The degrade factors after wildfires are 0.99 after 1 year, 0.89 after 2 years, 0.58 after 3 years [51], 0.22 after 4 years, and 0 after 5 years and later [16]. Using elasticities from various sources and a discount rate of 4%, the producer of damaged timber was estimated to lose a welfare of USD 51.5 million due to the lower log price, whereas the consumer lost a welfare of USD 79.7 million due to the

lower volume from the burned sites in the following years and the higher price from the decreased inventory [24]. Additionally, the timber price within the fire zone decreased by 28.7% in 2004 and 22.3% in 2005, whereas the stumpage price in the regional market outside the fire zone decreased by 10.7% in 2004 and 4.3% in 2005.

Prestemon and Holmes (2010) presented a conceptual model to estimate the economic impacts of hurricanes on timber producers and consumers, offer a framework to estimate welfare impacts, and illustrate the advantage of a welfare theoretic model by example [49]. The volume of damaged timber from six severe hurricanes, Camille in 1969, Hugo in 1989, Frances and Ivan in 2004, and Katrina and Rita in 2005, are measured, and the economic surplus for producers and consumers in 2005 dollars are quantified. Based on the economic estimates, they suggest that timberland investors could minimize their economic losses by diversifying their holdings geographically, which suggests that they could favor areas far from hurricane storms but still benefit from market-level price enhancements.

Spruce budworm (SBW) (Choristoneura fumiferana) and forest tent caterpillar (FTC) (Malacosoma disstria) are two widely spread forest insects in New Brunswick (NB) and Saskatchewan (SK), Canada. Chang et al. (2011) estimate the social benefits of SBW and FTC controls by surveying households' willingness-to-pay [50]. Their results suggest that the social benefit (welfare gains) in NB amounts to CAD 14.3–20.8 million for SBW control and CAD 7.9–14.5 million for FTC control. Additionally, the social benefit (welfare gains) in SK is CAD 22.2–32.4 million for SBW control and CAD 11.7–22.0 million for FTC control.

Henderson et al. (2022) simulated the damage to the forest growing stock and forest area, relying on forest inventory and remote sensed products, and examined the impacts of Hurricane Michael on markets and welfare [51]. The affected areas include the Florida Panhandle and adjacent areas of Alabama and Georgia. They find that timber prices decreased initially, rose once the demand returned to normal, and finally converged to the baseline trend. Compared with the no-hurricane scenario, the welfare gain for pine sawtimber producers increased by 1.2 to 1.5 times, and the welfare for consumer decreased by 0.6 to 0.8 times. The hardwood sawtimber producers gained 1.8 times the welfare, and the welfare for consumers decreased to half.

## 4. Discussion and Conclusions

To the best of our knowledge, this paper is the first paper that reviews the economic impacts of forest disturbances on the forest product market and summarizes four common models depending on the perspective of the economic analysis. To select the relevant papers in this topic, we exploited the Web of Science (WOS) database and finally kept 25 highly relevant papers focusing on loss estimation, impacts on economic sectors, price dynamics, and welfare redistribution. This literature review is important to provide useful information that contributes to the understanding of models of the economic impacts of forest disturbances, especially within the context of intensifying climate change and increasing forest disturbances. This literature review leads to a number of conclusions below.

The "with and without" analysis for economic loss evaluation is the most straightforward approach to estimate the economic losses of forest disturbances. It simply assumes that the timber price is constant over time and that the salvage logging is not substantial enough to cause price dynamics. The traditional forest damage appraisal approach measures producers' or policymakers' willingness-to-pay to avoid the forest damage. Though this approach is relatively simple, it is easy to compute, and it provides participants with an approximations of economic losses instantly. Although the limitation of this analysis is that the evaluation is roughly primary, this analysis is valuable for the first response to unexpected forest disturbances.

The general equilibrium model or partial equilibrium model allow individuals to analyze the impact of forest disturbances across all economic sectors or selected economic sectors of interest. In a general equilibrium model or partial equilibrium model with several markets, the equilibrium prices and quantities are endogenous, which indicates that there are feedback effects within markets. Compared with the "with and without" analysis,

the feedback effect in the equilibrium models would generate more accurate estimates of economic welfare. This model is especially helpful for policymakers because the impact of forest disturbances on macroeconomic activities can be examined. However, there are some limitations in the general or partial equilibrium models. Firstly, these models are subject to a lack of empirical support for model parameterization and rely on ad-hoc parameters [52,53]. Secondly, the model often ignores individuals' heterogeneous preferences and behavior [54–56]. Moreover, the forest sector is highly aggregated in the GE framework [57,58]. The application of the equilibrium models for forest disturbances should be modified to address these issues [59].

The intervention model allows researchers to conduct a comprehensive and sound outcome analysis of large forest disturbances. Salvage logging usually has a substantial impact on the current markets but minimal impacts on the long run markets, because the salvaged timber accounts for a large percentage in the annual allowable cut but a small percentage in the standing inventory. The intervention model is able to distinguish between the disturbance stage, salvage logging stage, and after-salvage-logging stage, and hence, the price dynamics during different stages of the forest disturbance can be estimated. The estimation of price dynamics could alter forest harvest and investment decisions through dynamic market adjustments.

The social welfare model allows for welfare estimation for different market participants, including landowners with damaged forests, landowners with undamaged forests, and consumers. Compared with the equilibrium models which can only estimate the welfare for a sector as a whole, the social welfare model can estimate the welfare changes for producers with damaged forests, producers with undamaged forests, and consumers in the forestry sector. Compared with the intervention model, the social welfare model also measures the short-run and long-run welfare impact. Therefore, the social welfare model can disaggregate the market participants and estimate the welfare dynamics over time.

This paper searched for the 25 most relevant papers in this topic, which might not be extensive. The obvious characteristics of forest disturbances are their rarity and large scale. The extreme events that cause significant economic losses in the forest are even less frequent. Therefore, there were not many papers we could refer to. Another reason is that the economic analysis of forest disturbances has not received enough attention. As pointed by Holmes (1991), an infestation at the large scale could cause economic losses of tens of millions of dollars. Therefore, a more comprehensive economic analysis for forest disturbances at the large scale should be performed.

Several extensions could be made to this literature review. We confine the economic models to the current most common models because we focus on the economic impacts of forest disturbances on the market and welfare. Future research could broaden the economic scope of this research and include the impact of forest disturbances on other aspects such as the residential property values. It could be interesting to extend the effects of forest disturbances on individuals' welfare or willingness-to-pay to avoid the catastrophic forest disturbances, which can be realized through the life satisfaction model. This could lead to the analysis of economic impacts of forest disturbance on individuals' welfare.

**Author Contributions:** Conceptualization, J.Z.; methodology, J.Z.; validation, Z.N.; investigation, J.Z.; writing—original draft preparation, J.Z.; writing—review and editing, J.Z., Z.N.; supervision, Z.N. All authors have read and agreed to the published version of the manuscript.

**Funding:** This study was supported by the National Natural Science Foundation of China (Program No. 72203095), and the Social Science Foundation of Jiangsu Province (Program No. 22EYC007).

**Institutional Review Board Statement:** Not applicable.

**Informed Consent Statement:** Not applicable.

**Data Availability Statement:** Not applicable.

**Acknowledgments:** We would like to thank all the authors and reviewers for their great guidance and help in writing this manuscript.

**Conflicts of Interest:** The authors declare no conflict of interest.

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
