# Peer review of "Models for the Economic Impacts of Forest Disturbances: A Systematic Review"

_land, doi:10.3390/land11091608_

Round 1

Reviewer 1 Report

This is an interesting manuscript reviewing "four economic impact models of forest disturbance", which are not novel but acceptable.It should be noted that there are several items in this manuscript that need to be corrected. Please modify the manuscript accordingly. The details are as follows:

1. If we regard the manuscript as a study on the economic impact model of forest disturbance, it should at least state the research conclusion.However, although the summary part discusses the key contents of this article, it is regrettable that the description of the conclusions obtained by comparing the advantages and disadvantages of the four models and the shortcomings of the existing research are obviously missing in the manuscript. Therefore, it is suggested to supplement them in the summary part.

2. In the introduction, it is suggested to quote 1-2 articles from 2021 to 2022.

3. In the part of "materials and methods", this paper explores the economic impact model of forest disturbance by searching the relevant literature in the database of Science Network (WOS). Although the collection time of the article is stated, it lacks the starting time. It is suggested to change it to the time range so that readers can read it clearly.Secondly, it should be noted that table 1 is the supporting evidence for which words, and that "the forest disturbance categories considered in the study include (1) - (6)" in the text discussion do not correspond to table 1. It is suggested to add a column of serial numbers in Table 1 or delete the serial numbers in the text discussion.At the same time, as for the statement "we have collected 963 papers on the economic impact of forest disturbance and manually selected 200 articles in the economic impact assessment", which is too subjective to read, it is suggested to supplement the theoretical basis or reasons for "manually selecting 100 articles".In addition, in the first paragraph of "materials and methods" and figure 1, 25 highly relevant papers were finally selected. In the second paragraph, "it can be seen from table 2 that 5 of the 36 papers were published in forest policy and economics", and in the discussion and conclusion part, "25 highly relevant papers were finally retained". In many places, the caliber is not uniform, and table 2 shows the details of 27 papers, which cannot correspond to 36 of the papers.

4. In the general part of Part III "model", it is suggested to supplement the official or authoritative definition of forest disturbance.

5. In Table 3, table 4 and table 5, pay attention to the time series of the cited documents. It is suggested to rearrange the authors and their corresponding text parts according to the time sequence. Meanwhile, it is suggested to place the quoted corner marks in the main discovery part.

6. The "model" in the third part of the article is subdivided into four branches, but these four branches only list the relevant literature of the model, and do not see what conclusions are drawn after combing the relevant literature. Therefore, it is suggested to supplement and improve them.As can be added in part 3.1, through the discussion of infection with MPB, foreign invasion pathogens, SOD, etc., we can find out which disaster has the greatest economic loss.

7. In the first paragraph of Table 4, "the partial equilibrium model only includes specific sectors, while the partial equilibrium model includes such as the global forest product model (GFPm) and the French forest sector model (ffsm) and the following" GFPm is a partial equilibrium model "

The statement is repeated. It is suggested to refine this paragraph.At the same time, the title formats of Table 4, table 5 and table 6 are not consistent with those of other tables. The format needs to be further standardized.

8. A transitional sentence is missing between the second and third paragraphs of part 3.3.The second paragraph discusses that the price fluctuation after forest disturbance depends on six factors. The third paragraph discusses the demand curve and the supply curve. The third paragraph aims to highlight whether the price elasticity of demand is the main reason that affects the price fluctuation after forest disturbance? It is suggested to make transition supplement.

9. 3.4 although there are many discussions on some social welfare models, they are only discussed from the perspectives of "hurricanes" and "fires". Considering that the probability of "diseases and pests" in real life is greater and more common than other perspectives, and the social welfare model of "diseases and pests" is also the concern of many scholars, it is suggested to replace it appropriately so that readers can understand the social welfare model from many aspects.

Author Response

Please find my response to the comments and suggestions in the attachment. Thank you so much for giving these great ideas.

Reviewer 2 Report

More evidence indicates that climate change may increase the frequency and range of forest disturbances. This manuscript reviews the most relevant studies investigating the economic impacts of forest disturbances. The models covered in the manuscript are particularly useful in understanding methodologies for economic impacts. The text is well written, and the research objectives are clear.  Overall, the manuscript is in good shape.  However, I have two concerns.  

1, As a review paper, the manuscript did a poor job of criticizing and comparing the pros and cons of different modeling frameworks. For example,  Section 3.2. discussed the equilibrium models. In general, CGE models are subject to a lack of empirical support for model parameterization and reliance on ad-hoc parameters. Secondly, the model often ignores individuals’ heterogeneous preferences and behavior. Moreover, the forest sector is highly aggregated in the GE framework. Fan and Davlasheridze (2019) have addressed this issue which is missing in this manuscript. The same thing applies to the partial equilibrium model. 

Fan, Q., Davlasheridze, M., 2019. Economic impacts of migration and brain drain after major catastrophe: The case of Hurricane Katrina. Climate Change Economics, 10(01), p.1950004.

Besides CGE models, PE models, GTM, GLOBIOM, and FASOM, have been widely used to address forest disturbance (uncertainty) as well. Examples include:

Reyer, C.P., Bathgate, S., Blennow, K., Borges, J.G., Bugmann, H., Delzon, S., Faias, S.P., Garcia-Gonzalo, J., Gardiner, B., Gonzalez-Olabarria, J.R., Gracia, C., 2017. Are forest disturbances amplifying or canceling out climate change-induced productivity changes in European forests?. Environmental Research Letters, 12(3), p.034027.

Prestemon, J.P., Turner, J.A., Buongiorno, J., Zhu, S., Li, R., 2008. Some timber product market and trade implications of an invasive defoliator: The case of Asian Lymantria in the United States. Journal of Forestry, 106(8), pp.409-415.

Wear, D.N., Murray, B.C., 2004. Federal timber restrictions, interregional spillovers, and the impact on US softwood markets. Journal of Environmental Economics and Management, 47(2), pp.307-330.

2. The references are a little bit inadequate. There are no reference articles regarding dynamic vegetation models. I would like to suggest that the authors at least add the following citations in the manuscript and give an in-depth analysis of this type of model.

Author Response

Please find my response to the comments. Thank you so much for the great suggestions. 

Round 2

Reviewer 2 Report

The author(s) has addressed most of my comments and I think the article makes a novel and positive contribution to the literature.